# Data Security Protocol with Blind Factor in Cloud Environment

**Ping Zhang [1], Huanhuan Chi [1,*], Jiechang Wang [2] and Youlin Shang [1]**

1   School of Mathematics and Statistics, Henan University of Science and Technology, Luoyang 471003, China;
    zping@haust.edu.cn (P.Z.); ylshang@haust.edu.cn (Y.S.)
2   Computer Teaching and Research Office, Physical Education College of Zhengzhou University,
    Zhengzhou 450044, China; 00352@peczzu.edu.cn
*   Correspondence: 190319100414@stu.haust.edu.cn; Tel.: +86-184-3860-5335

**Abstract:** Compared with the traditional system, cloud storage users have no direct control over their data, so users are most concerned about security for their data stored in the cloud. One security requirement is to resolve any threats from semi-trusted key third party managers. The proposed data security for cloud environment with semi-trusted third party (DaSCE) protocol has solved the security threat of key managers to some extent but has not achieved positive results. Based on this, this paper proposes a semi-trusted third-party data security protocol (ADSS), which can effectively remove this security threat by adding time stamp and blind factor to prevent key managers and intermediaries from intercepting and decrypting user data. Moreover, the ADSS protocol is proved to provide indistinguishable security under a chosen ciphertext attack. Finally, the performance evaluation and simulation of the protocol show that the ADSS security is greater than DaSCE, and the amount of time needed is lower than DaSCE.

**Keywords:** cloud computing environment; semi-trusted third party; user file upload; user file download; data security

## 1. Introduction

Cloud computing is expected to be the next generation of IT enterprise architecture. It is one of the best choices for big data processing and analysis, allowing users to remotely store and analyze their data with shared computing resources [1]. With the rapid growth in user data scale, cross-user cloud storage has become the mainstream application form for data storage; from simple backup systems to cloud storage systems, users can use low-cost, scalable online services [2]. Users outsource data to the cloud server, which performs data storage and management. This form of application fundamentally changes the way resources are deployed and services are provided, avoiding the heavy costs of local hardware maintenance [3].

At present, in the scenario of data storage encryption hosted by a third party, the common products are: Ali ESC cloud disk encryption, Tencent data encryption service CloudHSM, etc., which have the advantages of minor changes, minor expenses, being suitable for large-scale data storage, and remote reading [4]. Cloud computing has many advantages, but it also faces some problems and challenges, such as the security, performance, and quality of the cloud, mentioned in the literature [5–7].

A cloud computing environment means that users will work within the network environment. User data security is restricted by the level of service technology provided by cloud computing service providers, and users themselves also affect the security of the cloud computing environment [8]. The potential of cloud services has yet to be fully realized due to user concerns about the security and privacy of their data in cloud services. These concerns are primarily about cloud operators reducing access to sensitive data, making cloud computing less acceptable in many areas, such as the financial sector and with government agencies. Cloud providers and tenants may be untrusted entities attempting

to tamper with or compute data storage [9,10]. These threats to data security have spurred the need to use encryption to achieve cloud computing security goals.

Encryption technology provides an alternative method to ensure data privacy and confidentiality. However, in cases with encryption, key management becomes the primary issue [11]. Therefore, in the cloud environment, it is imperative to put forward a protocol that can guarantee user data security.

In 2019, Wu and Ling [12] proposed an improved cloud storage data integrity verification method, using bilinear to verify the data integrity of the technology to achieve an open verification function, and they designed an index table mechanism for dynamic verification. However, this method does not introduce the key manager and does not encrypt the files uploaded to the cloud storage.

To isolate user data information from user identity information, Zhan and Nie [13] proposed a cloud storage architecture protocol based on trusted third parties, which realized service quality evaluations for cloud storage providers to trusted third parties and used quality evaluation systems of trusted third parties to evaluate cloud storage providers. He et al. [14] proposed a data security protocol for trusted third-party platforms based on RSA one-time keys. RSA one-time key technology is used to realize the functions of secure encryption data. Then, one-time key generation is managed by a trusted third-party platform. Qian and Xie [15] proposed a CP-ABE cloud storage access control protocol based on trusted third parties. Based on the data block, the protocol effectively solves problems in data security, client key management and distribution, and excessive loads by introducing a trusted third party and uses CP-ABE mechanisms to ensure secure access control. To solve the problem of data sharing security in the multicloud storage system (MC-SS), Zhou et al. [16] designed an attribute mapping mechanism, which extended the attribute-based encryption based on ciphertext policy (CP-ABE) and proposed an ABE access control model with multi-authority CP to meet access control requirements for multicloud storage. However, in the real environment, access control protocols based on trusted third parties are ideal, and the protocols based on semi-trusted third parties are more practical and operable than the protocol based on trusted third parties.

Akhila et al. [17] proposed a data security system protocol based on a semi-trusted third parties in the cloud environment. The system provides key management, access controls, and file confirmation and deletion. The protocol uses the Shamir threshold secret sharing algorithm to manage the keys. Jin et al. [18] proposed BTDA, a semi-trusted third-party dynamic cloud data update audit protocol. The semi-trusted third party deals with update audits instead of users, so during the update audit process, the user can be offline, thereby reducing the communication costs and the computational costs on the user side. BTDA uses data blind and proxy re-signature technology to prevent semi-trusted third parties and cloud servers from obtaining sensitive user data. Tang et al. [19] designed and implemented file assure deletion (FADE) protocol, a secure overlay cloud storage system that achieves fine-grained, policy-based access control and assured file deletion. It associates outsourced files with file access policies, and assuredly deletes files to ensure they are unrecoverable by anyone upon revocations of file access policies. FADE is built upon a set of cryptographic key operations that are self-maintained by a quorum of key managers that are independent of third-party clouds. In addition, as an extension of FADE, Tang and other methods are still based on CP-ABE for access control. Ali et al. [20] considered that there is a man-in-the-middle attack between clients and key managers in FADE, so they added key exchanges and digital signatures, and proposed DaSCE, in which key managers are semi trusted third parties, and the system also provides key management, access controls, file guarantee deletion, and other functions. Reviewing the DaSCE for cloud environments with semi-trusted third parties proposed, in [21], although Ali analyzed some problems existing in the FADE protocol, they believed that the key manager was a semi-trusted third party and protected the man-in-the-middle attack between the client and key managers (KM), but it did not resolve the security threat from KM well (KM intercepts and decrypts the communication data between the client and cloud). Even in the case of

multiple key managers, if they conspire to attack, the threat still exists. Based on this, we propose a more secure protocol-ADSS.

## 2. Preliminaries

### 2.1. Indistinguishability

For $\prod = (\text{Gen}, \text{Enc}, \text{Dec})$, test $\text{Pr}iv_{A,\prod}^{eav}(n)$ of PPT adversary $A$ is defined below:

1.  Adversary $A$ input $1^n$, output a pair of messages $m_0, m_1$ of the same length.
2.  Run $\text{Gen}(1^n)$ to generate a key $k$, select a random bit $b$, $b \leftarrow \{0,1\}$, ciphertext $c \leftarrow \text{Enc}_k(m_b)$ is computed and given $A$, $c$ is the challenge ciphertext.
3.  $A$ outputs a bit $b'$, $b' \in \{0,1\}$.
4.  If $b = b'$ output 1, otherwise output 0.
5.  If $\text{Pr}iv_{A,\prod}^{eav}(n) = 1$, it means success.

A private key encryption protocol $\prod$ is indistinguishable from eavesdropping adversaries. For any PPT adversary $A$, there exists a negligible function $negl(n)$, such that:

$$\text{Pr}\left[\text{Pr}iv_{A,\prod}^{eav}(n) = 1\right] \leq \frac{1}{2} + \text{neg1}(n)$$

### 2.2. Indistinguishability of Chosen Ciphertext Attack

The test $\text{Pr}iv_{A,\prod}^{\text{cca}}(n)$ is defined as follows:

1.  Key generation: $k \leftarrow \text{Gen}(1^n)$.
2.  Adversary $A$ input $1^n$, using the oracle $\text{Enc}_k$ and $\text{Dec}_k$, output two messages of equal length $m_0, m_1$.
3.  Elect a random bit $b$, $b \leftarrow \{0,1\}$, let $c := \text{Enc}_k(m_b)$, send $c$ to adversary $A$.
4.  Adversary $A$ continues to use oracle $\text{Enc}_k$ and $\text{Dec}_k$. Restriction: Cannot query the plaintext of ciphertext $c$. Output a bit $b' \in \{0,1\}$.
5.  Use output: If $b = b'$ output 1, otherwise output 0.

If $\text{Pr}iv_{A,\prod}^{cca}(n) = 1$, then $A$ is successful.

A private key encryption protocol $\prod$ has indistinguishable encryption under the chosen ciphertext attack (CCA), for any PPT adversary $A$, there exists a negligible function $negl(n)$, such that:

$$\text{Pr}\left[\text{Pr}iv_{A,\prod}^{\text{cca}}(n) = 1\right] \leq \frac{1}{2} + \text{neg1}(n)$$

### 2.3. Large Integer Factorization

Large integer factorization problem (IF problem): Given odd complex number $N$, solve its prime factorization $N = p_1^{e_1} p_2^{e_2} \cdots p_r^{e_r}$, where $p_i$ is the distinct prime number, $e_i$ is the number of $p_i$ and $e_i \geq 1$.

Large integer factorization difficult hypothesis (IF hypothesis): An integer resolver is a PPT algorithm $A$, which satisfies the probability $\omega > 0 : w = \text{Prob}[L(N)|N, 1 < L(N) < N]$. Let IG be an integer generator, input $1^\lambda$, and output $N = pq$ of $2\lambda$ bit in polynomial time of $\lambda$, where $p$ and $q$ are random odd prime numbers of $\lambda$ bits. For all sufficiently large $\lambda$, there is no large integer factorization algorithm generated by $IG(1^\lambda)$.

### 2.4. FADE Security

In FADE [19], the symbols and their meanings are used (see Table 1), and $K$ and $S_i$ are random symmetric keys generated by the client. In the file upload phase, the client sends a policy file $P_i$ to KM; KM generates private key $(d_i, n_i)$ (secret preservation) and public key $(e_i, n_i)$ (sent to client) associated with $P_i$; the client encrypts $S_i$ to obtain $S_i^{e_i} \text{mod} n_i$, and then $S_i$ encrypts $K$ to get $\{K\}_{S_i}$. After that, the client will upload $P_i, \{F\}_K, \{K\}_{S_i}, S_i^{e_i} \text{mod} n_i$ to the cloud, and the client finally clears the local keys and files. For the sake of simplicity, we will omit "$\text{mod} n_i$" in the discussion. In the file download phase, after downloading the file and encryption key from the cloud, the client generates a random value $R$ as the

blinding factor and calculates $R^{e_i}$, multiplies it by $S_i^{e_i}$ to obtain $(S_iR)^{e_i}$, and sends $(S_iR)^{e_i}$ to the key manager KM to decrypt. KM decrypts $(S_iR)^{e_i}$ with $d_i$ and returns $S_iR$ to the client. The client decomposes $S_i$ from $S_iR$, and decrypts $K$, and finally decrypts $F$. The aforementioned is the file upload and download situation of a single key manager, and a case of multiple key managers will not be repeated.

**Table 1.** Symbols and meanings.

| Symbol | Meaning |
|---|---|
| $F$ | Data file |
| $K$ | A symmetric key, data key |
| $P_i$ | Policy file |
| $P_j$ | Forged strategic files |
| $(e_i, n_i)$ | KM generated public key parameters |
| $(d_i, n_i)$ | Public private key pair |
| $\{\}_{KEY}$ | Encryption with symmetric key |
| $S_i$ | A symmetric key corresponding to $P_i$ |

Ali [16] believes that when there is an intruder attack between the client and KM in the file upload phase of the FADE protocol (see Figure 1), the intermediary can intercept $P_i$ and send $P_j$ (forged $P_i$) to KM, and then KM sends $(e_i, n_i)$. The intermediary intercepts $(e_i, n_i)$ and sends the forged parameter $(e_j, n_j)$ to the client. The client uses the $(e_j, n_j)$ encryption key and uploads to the cloud, and the client cannot determine whether the $(e_j, n_j)$ received is from KM or other parties.

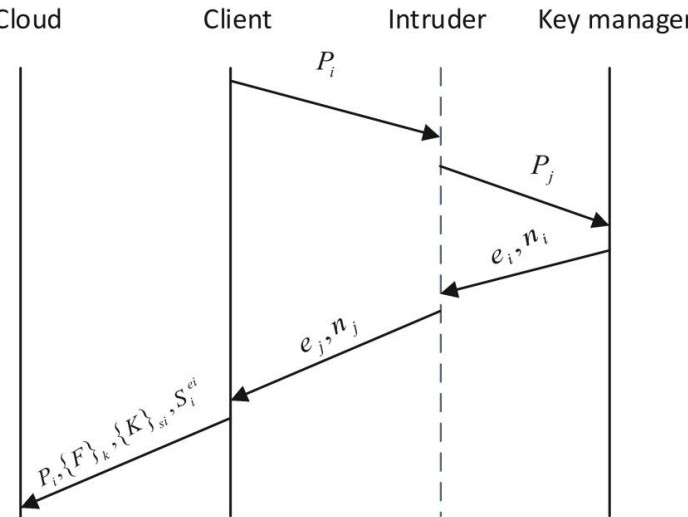

**Figure 1.** Man-in-the-middle attack during file upload.

In the file download stage, the intermediary can use its private key $(d_j, n_j)$ to intercept and decrypt the data. Similarly, in cases of multiple key managers, upload and download also face the same security problems.

*2.5. DaSCE Protocol*

2.5.1. DaSCE File Upload

To determine a session key, Ali assumes that parameters $\alpha$ and $p$ are fixed and open to all parties, where $\alpha$ is a large number as the primitive root and $p$ is a big prime number. The entire process consists of the following steps:

1. The client generates a random number $x$ and calculates $\alpha^x \bmod p$, and sends it to KM.
2. KM generates a random number $y$ and computes $\alpha^y \bmod p$. KM also computes $(\alpha^x)^y$ as the session key $K$ between him and the client.
3. KM generates $\{\alpha^y, \alpha^x\}$ digital signature $(S_{KM}\{\alpha^y, \alpha^x\})$ and uses the session key to generate encryption $E_k(S_{KM}\{\alpha^y, \alpha^x\})$.
4. KM sends $(\alpha^y, E_k(S_{KM}\{\alpha^y, \alpha^x\}))$ to the client.
5. The client first computes the session key $K = (\alpha^y)^x$, and declassifies $E_k(S_{KM}\{\alpha^y, \alpha^x\})$, then verifies the signature.
6. The client calculates $E_k(S_{Cli}\{\alpha^x, \alpha^y\})$ and $E_k(P_i)$, and sends them to KM.
7. KM verifies the digital signature of the client, after which KM declassified $P_i$ and generates $(e_i, n_i)$ related to $P_i$ and saves $P_i$.
8. KM calculates $E_k(e_i, n_i)$ and sends it to the client.
9. The client encrypts the file $F$ with the data key $K$, computes the MAC with $IK$ (to verify the integrity of $F$), $S_i$ encrypted $K$ and $IK$, then uses $e_i$ to encrypt $S_i$, and the client uploads the encrypted data to the cloud.
10. The client deletes all keys except the public key parameters sent by KM.

The file upload process can be seen in Figure 2. For simplicity, the $\bmod p$ used in calculating the session key is omitted.

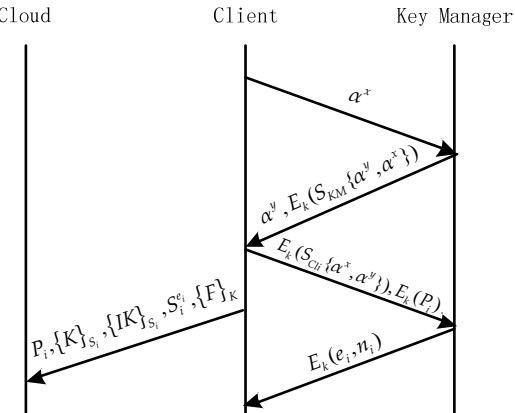

**Figure 2.** Single key manager DaSCE file upload.

The multi-key managers file upload, according to the Shamir $(k, N)$ threshold secret sharing algorithms, can be seen in Figure 3. $S_i$ is divided into $N$ shares, and each $N$ KM generates a pair of public and private keys.

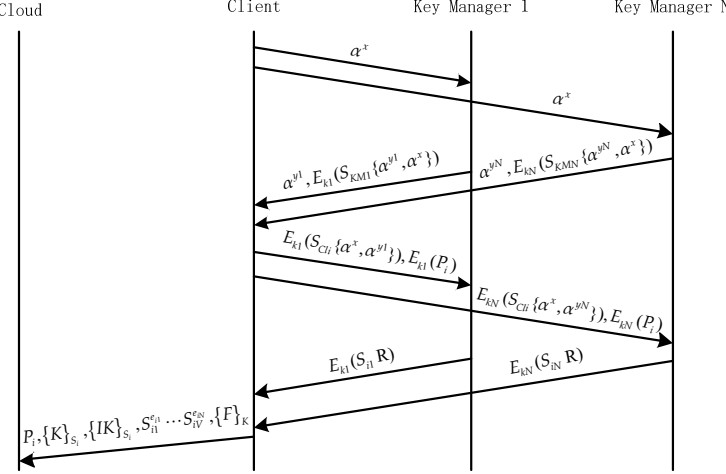

**Figure 3.** Multi-key manager DaSCE fileupload.

### 2.5.2. DaSCE File Download

The DaSCE single key manager file download process is similar to FADE, but to prevent man-in-the-middle attacks, the session key should be established before the client and KM, and then encrypted by the key.

DaSCE file downloads of multi-key managers (see Figure 4). After downloading ciphertext from the cloud, the client determines the session key with $N$ KM, it selects a random number $R$ and performs $S_{i1}^{ei1} R^{ei1}, \cdots, S_{iN}^{eiN} R^{eiN}$ operation, then separately sends them to $N$ KM decrypts. The client extracts $S_i$ from the received $S_i R$. According to the Shamir $(k, N)$ threshold secret sharing algorithm, $S_i$ can be generated by at least $K$ copies of $S_{iS}$, and finally decrypts the file $F$.

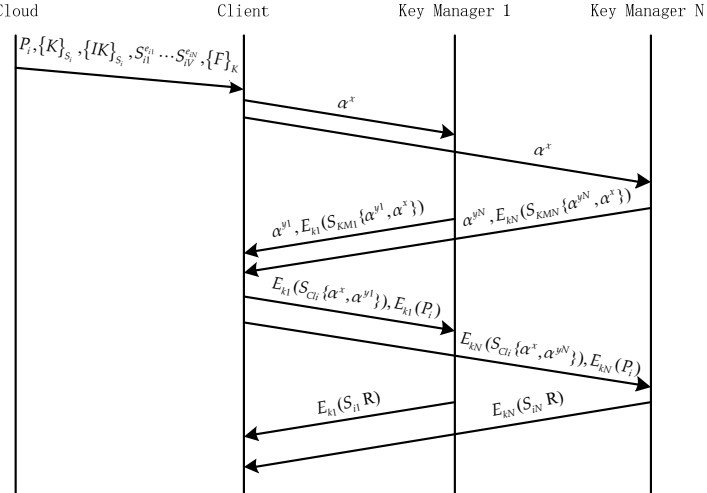

**Figure 4.** Multi-key manager DaSCE file download.

## 3. ADSS Model Definition

### 3.1. System Model

The system model for this paper (see Figure 5) includes the following entities: User (US), (single or multiple) KM, and the Cloud. Considering that the user may change the client (so use US instead of Client), save local storage space, and avoid information disclosure due to attacks, users will clear a large number of local keys and files after uploading data to the cloud. To share the security risks, restrict the cloud, and save computing resources, the user US connects with the key manager KM, which is the entity managing the key certificates in the network. It can provide high-performance computing services and can quickly encrypt or decrypt data for users. The general process for the model is below:

1.  The user encrypts the data by using the public key provided by the key manager KM, and then uploads the ciphertext to the cloud, then clears a large number of local keys and files, and only stores the blind factor and associated information in its USB-key (UKey).
2.  After downloading the ciphertext from the cloud, the user transmits some ciphertext to KM for decryption, and then the user decrypts the plain text by using its blind factor.

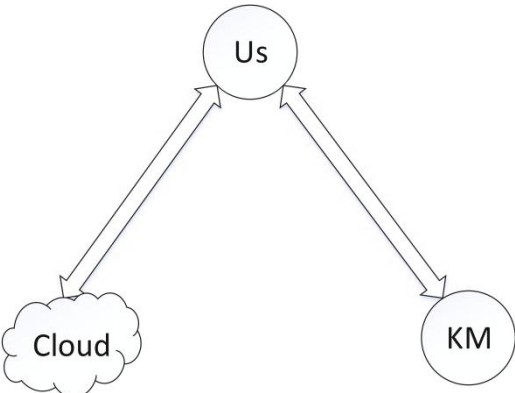

**Figure 5.** ADSS system model.

### 3.2. Security Model

In ADSS, KM is semi-trusted. It may launch an active attack on the communication between users and the cloud to intercept and decrypt the data uploaded or downloaded by users. Of course, a middleman can launch the same attack. In cases of multiple key managers, it is also possible to intercept and decrypt user data if the key managers conspire to attack. In the ADSS security model, the KM or middleman is called attacker $A$, which requires that the nsew protocol can resist the attack from $A$. The indistinguishability under the chosen ciphertext attack (IND-CCA) security of the protocol is defined by the interactive game between attacker $A$ and challengers:

1. Initialization. Challenger generation system ADSS, adversary $A$ obtains the public key of ADSS.
2. Ask. Adversary $A$ makes a decryption inquiry to the challenger. After the challenger decrypts, he will give the plain text to adversary $A$.
3. Challenge. Adversary $A$ outputs two messages of the same length $m_0, m_1$, and then receives ciphertext $C_b$ from the challenger, where the random value $b \leftarrow \{0, 1\}$.
4. Guess. Adversary output $b'$, if $b' = b$, then the adversary $A$ attack is successful.

**Definition 1.** *If polynomial time Adversary $A$ breaks through the aforementioned security model with negligible advantage $Adv = \left| \Pr[b' = b] - \frac{1}{2} \right|$, then we say that the protocol proposed in this paper is IND-CCA security.*

$$Adv = \left| \Pr[b' = b] - \frac{1}{2} \right|$$

### 3.3. ADSS Protocol

To make up for the shortcomings of FADE and DaSCE protocols, completely eliminating the security threat of KM, we propose the ADSS protocol. $K_i$ is a random symmetric key generated by user Us, corresponding to $P_i$. Us encrypts file $F$ with data key $K_i$, and encrypts $K_i$ with public and private key pair $(e_i, n_i)$ generated by KM.

### 3.4. File Upload

When the data are uploaded to the cloud (see Figure 6), the user sends a policy file $P_i$ to KM, and it requests to generate a pair of public and private keys. KM generates a public-private key pair associated with $P_i$ and sends the public key $(e_i, n_i)$ to the user. Different from the DaSCE protocol, the user encrypts file $F_i$ with $K_i$ to generate $\{F_i\}_{K_i}$, and generates a random blinding factor $R_i$ with time stamp $t$, calculates $R_i^{e_i}$, and multiplies it by $K_i^{e_i}$ to obtain $(K_i R_i)^{e_i}$. After that, the user uploads $P_i, (K_i R_i)^{e_i}, \{F_i\}_{K_i}, t$ to the cloud.

Finally, the user clears all local keys and files and only stores the related policy file $P_i$, blinding factor $R_i$, and time stamp $t$ in his personal UKey.

$$Us \to KM : P_i$$
$$KM \to Us : (e_i, n_i)$$
$$Us : \{F_i\}_{K_i}, K_i{}^{e_i} \cdot R_i{}^{e_i}$$
$$Us \to Cloud : P_i, (K_i R_i)^{e_i}, \{F_i\}_{K_i}, t$$
$$Us\_UKey : save(P_i, t, R_i)$$

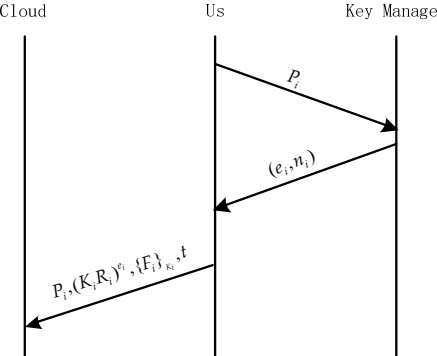

**Figure 6.** ADSS single KM file upload.

The case of multiple key managers (see Figure 7). The biggest difference from a single key manager is that: users use threshold secret sharing algorithm Shamir $(k, N)$ (where $1 \le b \le N$) to divide $K_i$ into $N$ shares of $K_{i1}, \cdots, K_{iN}$, and then blind encrypt them, respectively.

$$Us \to KM_1, \cdots, KM_N : P_i$$
$$KM_1, \cdots, KM_N \to Us : (e_{i1}, n_{i1}), \cdots, (e_{iN}, n_{iN})$$
$$Us : \{F_i\}_{K_i}, divide(K_i) = K_{i1}, \cdots, K_{iN}$$
$$Us : K_{i1}{}^{e_{i1}} \cdot R_i{}^{e_{i1}}, \cdots, K_{i1}{}^{e_{iN}} \cdot R_i{}^{e_{iN}}$$
$$Us \to Cloud : P_i, (K_{i1} R_i)^{e_{i1}}, \cdots, (K_{iN} R_i)^{e_{iN}}, \{F_i\}_{K_i}, t$$
$$Us\_UKey : save(P_i, t, R_i)$$

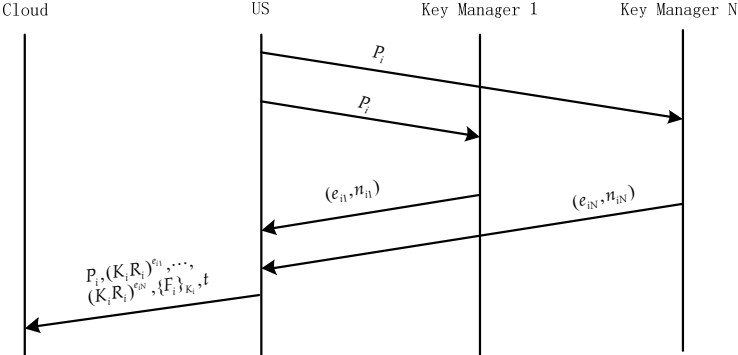

**Figure 7.** ADSS multi-KM file upload.

*3.5. File Download*

After downloading the file and encryption key from the cloud, the user sends $P_i, (K_i R_i)^{e_i}$ to the key manager KM for decryption. KM decrypts $(K_i R_i)^{e_i}$ with $d_i$ and returns $K_i R_i$ to the user. The user finds the corresponding blinding factor $R_i$ from its UKey through the

policy file $P_i$ and time stamp $t$, then decomposes $K_i$ from $K_i R_i$, and finally decrypts to get $F_i$. The specific process is shown in Figure 8.

$$Cloud \rightarrow Us : P_i, (K_i R_i)^{e_i}, \{F_i\}_{K_i}, t$$
$$Us \rightarrow KM : P_i, (K_i R_i)^{e_i}$$
$$KM \rightarrow Us : ((K_i R_i)^{e_i})^{d_i} = K_i R_i$$
$$Us\_UKey : find(P_i, t) = R_i$$
$$Us : K_i R_i / R_i = K_i,$$
$$\left\{ \{F_i\}_{K_i} \right\}_{K_i} = F_i$$

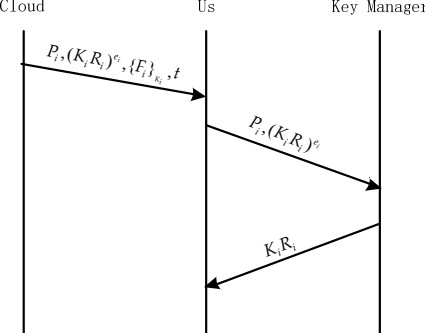

**Figure 8.** ADSS single-KM file download.

The case of multiple key managers (see Figure 9). Users download $P_i, (K_{i1} R_i)^{e_{i1}}, \cdots,$ $(K_{iN} R_i)^{e_{iN}}, \{F_i\}_{K_i}, t$ from the cloud and send $P_i, (K_{i1} R_i)^{e_{i1}}, \cdots, P_i, (K_{iN} R_i)^{e_{iN}}$ to $KM_1, \cdots,$ $KM_N$ to decrypt. $b$ key managers perform decryption and return $bK_{ii} R_i$ to the user, users find the corresponding blinding factor $R_i$ from their Ukey through the policy file $P_i$ and time stamp $t$, and then decompose $bK_{ii}$ from $bK_{ii} R_i$. Then, the user can recover $K_i$ from $K_{ii}, \cdots, K_{i,i+b-1}$ by Shamir $(k, N)$, and finally decrypt $\{F_i\}_{K_i}$ with $K_i$.

$$Cloud \rightarrow Us : P_i, (K_{i1} R_i)^{e_{i1}}, \cdots, (K_{iN} R_i)^{e_{iN}}, \{F_i\}_{K_i}, t$$
$$Us \rightarrow KM_1, \cdots, KM_N : P_i, (K_{i1} R_i)^{e_{i1}}, \cdots, P_i, (K_{iN} R_i)^{e_{iN}}$$
$$KM_1, \cdots, KM_N \rightarrow Us : K_{i1} R_i, \cdots, K_{iN} R_i$$
$$Us\_UKey : find(P_i, t) = R_i$$
$$Us : K_{ii} R_i / R_i = K_{ii}, \cdots, K_{i,i+b-1} R_i / R_i = K_{i,i+b-1}$$
$$US : Shamir(b, N)[K_{ii}, \cdots, K_{i,i+b-1}] = K_i$$
$$\left\{ \{F_i\}_{K_i} \right\}_{K_i} = F_i$$

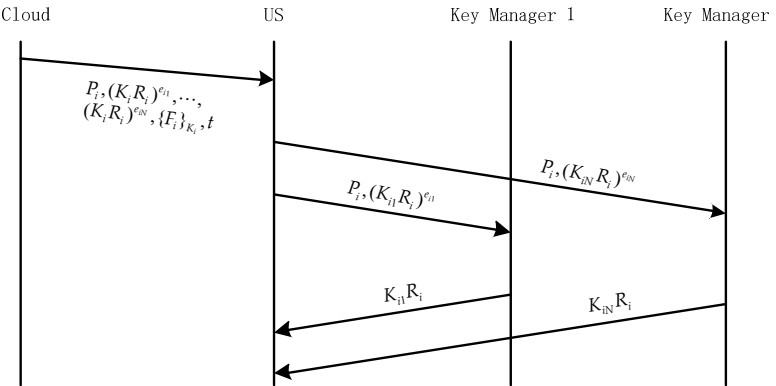

**Figure 9.** ADSS multi-KM file download.

## 4. Security Analysis

To prevent network sniffing attacks and security threats from the key manager, DaSCE does not add the blind factor $R$ before the user uploads the file. After downloading the file, the blind factor $R$ is added before sending $S_i^{e_i}$ to KM. Although this can prevent network sniffing attacks, it cannot prevent the KM from actively attacking the communication between users and the cloud to intercept and decrypt the data. To prevent man-in-the-middle attacks, Ali exchanged the key between the client and KM first and added a digital signature, but this measure still cannot prevent KM from intercepting $S_i^{e_i}$ and decrypting $S_i$ in advance. In cases of multiple key managers, it is also possible to intercept and decrypt user data if the key managers conspire to attack.

In this protocol, users add the blinding factor $R_i$ before uploading files. The specific operation is that the user first generates $R_i$ locally, calculates $(K_i R_i)^{e_i}$, and uploads it to cloud storage, along with other data. After that, when users communicate with the cloud (whether uploading or downloading files), only users know $R_i$; even if KM or middleman intercepts data, it is difficult to decompose $K_i$ by $K_i R_i$ ($K_i$ and $R_i$ are random large prime numbers) [22]. In the case of multiple key managers, if the key managers conspire to attack, they will encounter the same difficulty.

**Theorem 1.** *In the case of large integer factorization difficulties, the ADSS protocol is IND-CCA secure for semi-trusted third-party KM attacks or man-in-the-middle attacks.*

Specifically, if an IND-CCA adversary $A$ (KM or middleman) attacks ADSS with a non-negligible advantage $\varepsilon$, then there must be an adversary $B$ who can solve the IF problem with at least a non-negligible advantage $2\varepsilon$.

Prove:

First, we give the IND-CCA game of ADSS as follows:

Let $C = (C_1, C_2) = ((K_i R_i)^{e_i}, \{m_i\}_{K_i})$

Use $Exp_{ADSS,A}^{IND-CCA}$ to represent the IND-CCA game of ADSS, then:

1.  Run $GenADSS$ to generate $n_i, e_i, d_i, K_i, R_i$, where $n_i, e_i, d_i$ are known, and $K_i, R_i$ are unknown;
2.  Adversary $A$ obtains message $m_{i0}, m_{i1}$;
3.  Randomly select a bit $b \leftarrow \{0, 1\}$, let $C^* = ((K_i R_i)^{e_i}, \{m_{ib}\}_{K_i})$;
4.  Send $n_i, e_i, d_i, C^*$ to $A$, $A$ outputs $b'$.

Returns 1 if $b' = b$, 0 otherwise.

The adversary cannot decrypt the target ciphertext $C^*$. The advantage of adversary $A$ is defined as:

$$Adv_{ADSS,A}^{IND-CCA} = \left| \Pr[Exp_{ADSS,A}^{IND-CCA} = 1] - 1/2 \right|$$

The following proves that the ADSS protocol can be reduced to the IF (large integer factorization) problem.

Adversary $B$ knows that $(n_i, e_i, d_i, \hat{C}_1)$, using $A$ (attack ADSS) as a subroutine, executes the following process: the goal is to calculate $\hat{K}_i = \frac{(\hat{C}_1)^{d_i} \bmod n_i}{\hat{R}_i}$.

1.  Choose a random number $\hat{K}_i$ as a guess for $\frac{(\hat{C}_1)^{d_i} \bmod n_i}{\hat{R}_i}$ (but $B$ does not actually know $\hat{R}_i$), and give $(n_i, e_i, d_i)$ to $A$.
2.  $K_i$ asked: $B$ creates a list $L$, the element type is triple $(R_i, C_1, K_i)$, and the initial value is $(*, \hat{C}_1, K_i)$, where $*$ indicates that the value of the component is currently unknown.

    $A$ can ask $L$ at any time. Let $A$ query $K_i$, $B$ calculate $K_i = \frac{(C_1)^{d_i} \bmod n_i}{R_i}$ and make the following response:

    a.  If there is one item $(R_i, C_1, K_i)$ in $L$, answer with $K_i$.
    b.  If there is one item $(*, C_1, K_i)$ in $L$, answer with $K_i$ and replace $(*, C_1, K_i)$ with $(R_i, C_1, K_i)$ in $L$.

c. Otherwise, select a random number $K_i$, answer with $K_i$ and store $(R_i, C_1, K_i)$ in the table.

3. Decryption inquiry: When $A$ asks $B$ to ask $(\overline{C}_1, \overline{C}_2)$, $B$ responds below:

a. If there is a first term in $L$, and the second element is $\overline{C}_1$ (the term $(\overline{R}_i, \overline{C}_1, \overline{K}_i)$ or $(*, \overline{C}_1, \overline{K}_i)$), then $\{\overline{C}_2\}_{\overline{K}_i}$ is used to answer.

b. Otherwise, select a random number $\overline{K}_i$, answer with $\{\overline{C}_2\}_{\overline{K}_i}$ and store $(*, \overline{C}_1, \overline{K}_i)$ in $L$.

4. Challenge: $A$ output message $m_{i0}, m_{i1}$, $B$ random selection $b \leftarrow_R \{0,1\}$, calculate $\hat{C}_2 = \{m_{ib}\}_{\hat{K}_i}$ and answer $A$ with $(\hat{C}_1, \hat{C}_2)$. Continue to answer $A$'s $K_i$ query and decryption query ($A$ cannot query $(\hat{C}_1, \hat{C}_2)$).

5. Guess: $A$ output guesses $b'$, $B$ checks $L$, and if there are items $(\hat{R}_i, \hat{C}_1, \hat{K}_i)$, then output $\hat{R}_i$.

Let $D$ be the event: when $A$ asks for $\hat{K}_i$ (that is $\frac{(\hat{C}_1)^{d_i} \bmod n_i}{\hat{R}_i}$) in the simulation, $\hat{K}_i$ appears in $L$.

In the above attack, if $\hat{K}_i$ does not appear in $L$, then $A$ fails to obtain $\hat{K}_i$. According to the security of $\hat{C}_2 = \{m_{ib}\}_{\hat{K}_i'}$, the

$$\Pr[b' = b | \overline{D}] = \Pr[Exp_{ADSS,A}^{IND-CCA} = 1 | \overline{D}] = 1/2$$

where $\overline{D}$ is the complement event of $D$. From the definition of $A$ in a real attack, we can know that:

$$Adv_{ADSS,A}^{IND-CCA} = \left| \Pr[Exp_{ADSS,A}^{IND-CCA} = 1] - 1/2 \right| = \varepsilon$$

Because:

$$\begin{aligned}
&\Pr[Exp_{ADSS,A}^{IND-CCA} = 1] \\
&= \Pr[Exp_{ADSS,A}^{IND-CCA} = 1 | \overline{D}] \Pr[D] \\
&\quad + \Pr[Exp_{ADSS,A}^{IND-CCA} = 1 | D] \Pr[D] \\
&\leq \Pr[Exp_{ADSS,A}^{IND-CCA} = 1 | \overline{D}] \Pr[\overline{D}] + \Pr[D] \\
&= 1/2 \Pr[\overline{D}] + \Pr[D] \\
&= 1/2(1 - \Pr[D]) + \Pr[D] \\
&= 1/2 + 1/2 \Pr[D]
\end{aligned}$$

That is:

$$\varepsilon = \left| \Pr[Exp_{ADSS,A}^{IND-CCA} = 1] - \frac{1}{2} \right| \leq \frac{1}{2} \Pr[D] \Pr[D] \geq 2\varepsilon$$

Therefore, in the above simulation process, $\hat{R}_i$ appears in $L$ at least with the probability of $2\varepsilon$, $B$ checks the elements in $L$ one-by-one in step 5, so the probability of success of $B$ is equal to $\Pr[D]$; therefore, $B$ at least solves the IF problem with a non-negligible advantage $2\varepsilon$, which is obviously in contradiction with the difficulty of large integer factorization, so the advantage $\varepsilon$ of an IND-CCA adversary $A$ (KM or middleman) to break ADSS is negligible. Therefore, the ADSS protocol is IND-CCA secure, and the theorem is proved.

## 5. Performance Evaluation

### 5.1. Simulation Experiment

The protocol has been verified in some universities for simulation experiments, in which the performance parameters of the cloud server are: 600 MB bandwidth, 16-core CPU, 64 GB memory, 8 TB storage; the performance parameters for the KM server are: 32-core CPU, 128 GB memory, 1 TB storage. Two computers are used to simulate the user to upload and download. Both computers are desktop computers (4-core CPU, 8 GB memory, 500 GB storage). We select files with sizes of 1 KB, 3 KB, 10 KB, 30 KB, 100 KB, 300 KB, 1 MB, 3 MB and 10 MB, respectively, for simulation test. In the upload and download phase, the

time cost of ADSS and DaSCE protocols is shown in Tables 2 and 3, the unit of time cost is seconds.

**Table 2.** Time cost of ADSS and DaSCE protocols in file upload stage.

| File Size<br>Protocol | 1 KB | 3 KB | 10 KB | 30 KB | 100 KB | 300 KB | 1 MB | 3 MB | 10 MB |
|---|---|---|---|---|---|---|---|---|---|
| DaSCE | 0.217 | 0.238 | 0.249 | 0.250 | 0.455 | 0.560 | 1.078 | 4.989 | 7.238 |
| ADSS | 0.138 | 0.158 | 0.168 | 0.181 | 0.376 | 0.479 | 0.998 | 4.909 | 7.159 |

**Table 3.** Time cost of ADSS and DaSCE protocols in file download stage.

| File Size<br>Protocol | 1 KB | 3 KB | 10 KB | 30 KB | 100 KB | 300 KB | 1 MB | 3 MB | 10 MB |
|---|---|---|---|---|---|---|---|---|---|
| DaSCE | 0.212 | 0.265 | 0.324 | 0.683 | 0.456 | 1.135 | 1.149 | 11.049 | 19.059 |
| ADSS | 0.131 | 0.164 | 0.223 | 0.181 | 0.582 | 1.106 | 1.068 | 10.967 | 18.960 |

Figures 10 and 11 are the simulation charts we made with MATLAB. In the simulation, the horizontal axis is the file size, the unit is KB, and the scale value is 100, 101, 102, 103, 104; the vertical axis is the time cost with unit s, where the scale value in Figure 10 is 10−1, 100, 101, and the scale value in Figure 11 is 10−1, 100, 101, 102; From Figures 10 and 11, we can see that the time needed for ADSS is less than that of DaSCE.

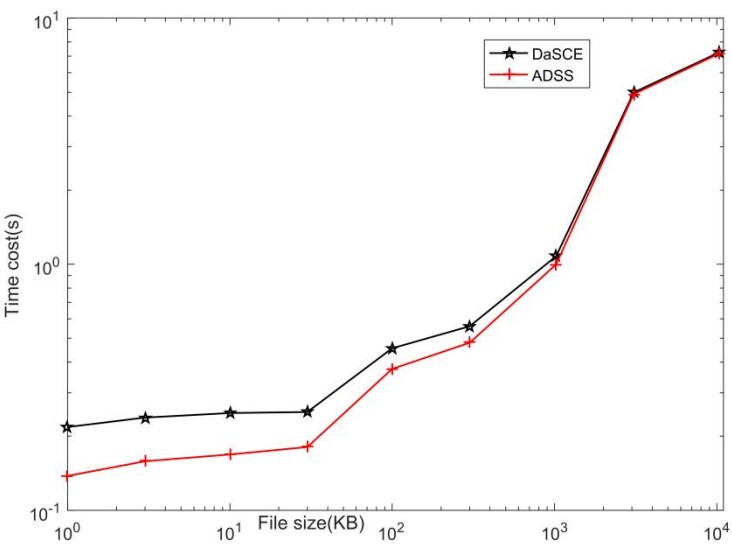

**Figure 10.** Comparison of file upload times in two protocols.

### 5.2. Performance Analysis

In the file upload stage, compared with DaSCE, this solution adds blinding calculation and UKey storage, eliminating key exchange (including digital signature) and one encryption calculation $\{K\}_{S_i}$, so the running time for this solution should be shorter than DaSCE at this stage.

In the file download stage, compared with DaSCE, this solution increases the user's reading from UKey, eliminating the need for blind calculations, key exchanges (including digital signature), and one-time encryption calculation $\{K\}_{S_i}$. Therefore, the running time of this solution at this stage should be longer than DaSCE is short.

In summary, the total running time for this program should be shorter than DaSCE.

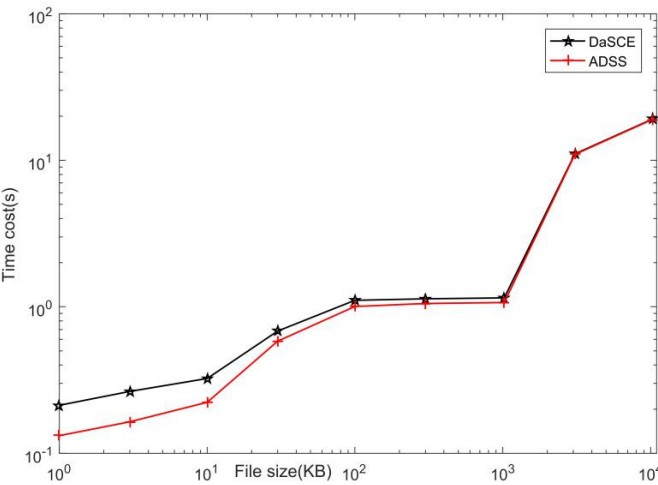

**Figure 11.** Comparison of file download times in two protocols.

## 6. Conclusions

Data security on the cloud affects the development of cloud technology applications. Reasonable and effective security algorithms and access control methods can improve user trust in cloud storage services, and the performance cost for the cloud storage system should also be considered. This paper fully considers security threats from the semi-trusted third-party KM and proposes an ADSS protocol. The analysis and simulation show that the security of this protocol is higher than that of DaSCE, and the running time is shorter than DaSCE, so it has higher practicality and operability.

**Author Contributions:** Conceptualization, P.Z. and H.C.; methodology, P.Z. and J.W.; software, Y.S.; validation, P.Z., H.C. and Y.S.; formal analysis, H.C.; investigation, H.C.; resources, P.Z.; data curation, J.W.; writing—original draft preparation, H.C.; writing—review and editing, P.Z.; visualization, P.Z.; funding acquisition, Y.S. All authors have read and agreed to the published version of the manuscript.

**Funding:** This research was funded by National Natural Science Foundation of China, grant number 12071112 and 11471102.

**Institutional Review Board Statement:** Not applicable.

**Informed Consent Statement:** Not applicable.

**Data Availability Statement:** Not applicable.

**Acknowledgments:** This work was supported by the National Natural Science Foundation of China, grant number 12071112 and 11471102.

**Conflicts of Interest:** The authors declare no conflict of interest.

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
