# Peer review of "Data Security Protocol with Blind Factor in Cloud Environment"

_information, doi:10.3390/info12090340_

Round 1

Reviewer 1 Report

The authors propose technologies to improve data security in the cloud when developing cloud applications and services. Algorithms and methods of access control are proposed that increase user confidence in cloud storage. At the same time, the cost of quality and performance of the cloud storage system is reduced. Considered are security threats emanating from destructive and malicious applications and ways to eliminate them based on the ADSS protocol, the security and performance of which is higher than DaSCE. Disadvantages that must be eliminated: 1) All drawings must be redone in a form that does not allow the intersection of letters and lines. 2) The lines must be orthogonal to each other. 3) All letters must be the same size, no compression or stretching.

Reviewer 2 Report

The paper addresses the security for the data stored in the cloud and it proposes a new  semi-trusted third-party data security protocol (ADSS) for solving the security threat of key manager.  It is based on adding time stamp and blind factor to prevent key managers and intermediaries from intercepting and decrypting user data.

Weakness:

1. The sentence “moreover, ADSS has shown to choose ciphertext attacks to be indistinguishability security under the conditions of large integer factorization problems“ must be more clear expressed

2. There are acronyms used in the paper that are not explained (eg. FADE, KM, IND-CCA, Ukey)

2. A description of the paper`s structure would be appropriate

3. I recommend introducing the enumerated steps in the figures (Figure 1, Figure 2, …  Figure 6, …)

4. The security model is not clearly described

5. The prove for Theorem 1 is not very clear explained

6. Performance Analysis need be more clearly explained. There are mistakes in expressing ideas and it is not clear what the authors want to express

7. The section for Simulation Experiment (Section 5.2) must be more descriptive
